# Intrathecal Synthesis Index of Specific Anti-*Treponema* IgG: a New Tool for the Diagnosis of Neurosyphilis

ⓘChloé Alberto,[a] Christine Deffert,[b] Nathalie Lambeng,[b] Gautier Breville,[c] Angèle Gayet-Ageron,[d] Patrice Lalive,[c] Laurence Toutous Trellu,[a] Lionel Fontao[a,b]

aDivision of Dermatology and Venereology, University Hospital of Geneva, Geneva, Switzerland
bDivision of Laboratory Medicine, University Hospital of Geneva, Geneva, Switzerland
cDivision of Neurology, University Hospital of Geneva, Geneva, Switzerland
dCRC & Division of Clinical Epidemiology, Department of Health and Community Medicine, University of Geneva & University Hospitals of Geneva, Geneva, Switzerland

**ABSTRACT** Neurosyphilis (NS) diagnosis is challenging because clinical signs are diverse and unspecific, and a sensitive and specific laboratory test is lacking. We tested the performance of an antibody index (AI) for intrathecal synthesis of specific anti-*Treponema* IgG by enzyme-linked immunosorbent assay (ELISA) for NS diagnosis. We conducted a retroprospective monocentric study including adults with neurological symptoms who had serum and cerebral spinal fluid (CSF) samples collected between 2006 and 2021. Two NS definitions were used. NS1 included patients with neurological symptoms, positive *Treponema pallidum* particle agglutination (TPPA) serology, and CSF-TPPA of ≥320, as well as CSF-leukocytes of >5 cells/mm³ and/or CSF-protein of >0.45 g/L and/or a reactive CSF-VDRL/RPR test. NS2 included patients with acute ocular and/or otologic symptoms, positive TPPA serology, and a response to NS treatment. Controls were patients with central nervous system disorders other than neurosyphilis. Anti-*Treponema pallidum* IgG were measured simultaneously in serum and CSF, and AI was calculated according to Reiber diagram. We assessed the AI test area under the curve (AUC), sensitivity/specificity, and estimated positive and negative predictive values. In total, 16 NS1 patients, 11 NS2 patients, and 71 controls were included. With an AI of ≥1.7 as a positive test for NS diagnostic, specificity was 98.6% (95% confidence interval [CI 95%] of 92.4 to 100.0) and sensitivity was 81.3% (CI 95% of 54.4 to 96.0) for NS1 and 98.6% (CI 95% 92.4 to 100.0) and 27.3% (CI 95% 6.0 to 61.0), respectively, for NS2. Positive and negative predictive values were >95% for NS1 and >85% for NS2, for prevalence above and below 20%. Measuring an AI for intrathecal synthesis of specific anti-*Treponema pallidum* IgG is a new promising tool highly specific for NS diagnosis.

**IMPORTANCE** In the context of a lack of a gold standard for the diagnosis of neurosyphilis due to either nonspecific or nonsensitive tests, we present in this article a new promising tool highly specific for NS diagnosis. This new test involves measuring an intrathecal synthesis index of specific anti-*Treponema* IgG by ELISA.

**KEYWORDS** neurosyphilis, intrathecal synthesis, specific anti-treponema IgG, otosyphilis, ocular syphilis

Syphilis, a sexually transmitted infection caused by the spirochete *Treponema pallidum*, occurs worldwide with a steadily increasing incidence. Neurosyphilis (NS) rate is thus expected to rise (1–4). At any stage of infection, the spirochete can invade both the central (meninges, brain, and spinal cord) and the peripheral nervous systems, leading to serious neurological sequelae in the absence of early treatment (5, 6). NS can be differentiated into an early stage that includes asymptomatic or symptomatic meningitis and meningovascular syphilis and a late stage that includes dementia paralytica and tabes dorsalis. Moreover, ocular syphilis and otologic syphilis can occur

Address correspondence to Chloé Alberto, chloe.alberto@unige.ch.

The authors declare no conflict of interest.

at any time of the disease but are more frequent during early NS. When the diagnosis of NS is considered, cerebrospinal fluid (CSF) examination is mandatory (7) to assess whether inflammation and/or intrathecal production of antibodies occurs in response to *Treponema pallidum* invasion. Unfortunately, there is currently not a simple sensitive and specific laboratory test to establish or exclude the diagnosis of NS. Indeed, high protein content and elevated white blood cells (WBC) in CSF are indicative of an inflammatory reaction in the CSF, but they are not specific markers for NS (7). Although a reactive non-*Treponema* test (VDRL/rapid plasma reagin [RPR]) in CSF associated with neurological symptoms is commonly used as diagnostic for NS, it is only 30% sensitive (2, 8–10). On the contrary, treponemal tests such as fluorescent treponemal antibody absorption (FTA-abs) and *Treponema pallidum* hemagglutination assay (TPHA)/*Treponema pallidum* particle agglutination (TPPA) in CSF are sensitive but suffer a poor specificity owing to the passive transfer of immunoglobulins across the blood-CSF barrier (4, 11, 12). Several tests, such as CSF-TPHA/TPPA index, which assess blood-meningeal barrier disruption, have been used to evaluate intrathecal synthesis of anti-treponemal antibody but are not yet validated for the diagnosis (10, 13). More recently, a TPPA titer higher than 320 or 640 was found to have a high specificity for NS (89 to 96%) but a low sensitivity (12 to 48%) (14, 15). To date, detection of *Treponema pallidum* DNA with PCR in CSF is not commonly used due to a very low sensitivity and a suboptimal specificity (8, 16–21).

Given these limitations, standardized definitions of NS have been established by the International Union against Sexually Transmitted Infections (8) and the Centers for Disease Control and Prevention (2, 7). Both used a combination of clinical symptoms or signs consistent with NS, serological evidence for syphilis, and abnormalities of the CSF (positive CSF-TPHA/TPPA and/or FTA test, as well as positive CSF-VDRL/RPR test, elevated CSF-WBC, or increased CSF-protein).

In the current diagnostic study, we assessed the diagnostic performance and the clinical utility of measuring an antibody index (AI) for intrathecal synthesis of specific anti-*Treponema* IgG for the diagnosis of NS. This AI can provide indirect evidence for *Treponema pallidum* invasion of the central nervous system (CNS) by demonstrating a production of local pathogen-specific antibodies. Normally, blood-brain barriers (BBB) restrict leakage of systemic antibodies into CSF, but their function changes in the presence of inflammation. Therefore, to discriminate between locally produced antibodies and systemic antibodies, a correction is needed to take the blood-CSF barrier function into account and correct for polyspecific antibody production in the brain.

## RESULTS

**Study population characteristics.** Twenty-seven patients with NS, 16 NS1 and 11 NS2, and 71 controls were recruited. Patients' characteristics are shown in Table 1. Patients with NS were not different from controls regarding age and HIV status. The high rate of HIV-positive patients in the controls is explained by their recruitment mainly from the infectious diseases department, which is a referral center for HIV infection. We observed a higher proportion of men among patients with NS than among controls ($P < 0.001$). Neurological, ocular, or hearing symptoms or a combination of symptoms were found, respectively, in 44.4% (12), 40.7% (11), 29.6% (8), and 7.4% (2) of patients.

**CSF analysis.** CSF of NS patients showed a significant increase of WBC compared to that of patients in the control group (Table 2). The predominant cells were lymphocytes (data not shown), and more than half (63%) contained plasma cells. CSF-proteins were slightly but significantly increased in NS patients compared to those in the control group. As illustrated by albumin quotient, we did not find any statistical difference of BBB disruption in NS compared to that in the controls. As expected, intrathecal IgG synthesis assessed by quantitative total IgG index and qualitative isoelectric focusing (IEF) was found more frequently in NS than in controls. This intrathecal synthesis was equally distributed between the type II ($n = 7$) and the type III ($n = 6$) oligoclonal bands (OCB) but was restricted to NS1 (Table 2). For NS1, specificity of the IEF was 73.2% (CI 95% of 61.4 to 83.1) and sensitivity was 81.3% (CI 95% of 54.4 to 96.0).

**TABLE 1** Study population characteristics

| Characteristic | Value for group | | | | P value[a] | P value[b] |
|---|---|---|---|---|---|---|
| | NS (n = 27) | NS1 (n = 16) | NS2 (n = 11) | Controls (n = 71) | | |
| Sex, n (%) | | | | | <0.001 | <0.001 |
| Women | 1 (3.7) | 1 (6.3) | 0 (0) | 34 (47.9) | | |
| Men | 26 (96.3) | 15 (93.7) | 11 (100) | 37 (52.1) | | |
| HIV seropositivity[c], n (%) | | | | | 0.157 | 0.377 |
| Negative | 20 (74.1) | 10 (62.5) | 10 (90.9) | 51 (82.3) | | |
| Positive | 7 (25.9) | 6 (37.5) | 1 (9.1) | 11 (17.7) | | |
| Mean age (±SD, p50: p25–p75) | 50.0 (±14.3, 51: 36–59) | 52.7 (±15.9, 55.5: 39.5–63.5) | 46.1 (±11.3, 49: 36–54) | 48.6 (±16.9, 47: 36–61) | 0.486 | 0.599 |

[a]Comparisons of proportions among the three groups.
[b]Comparison between patients with NS and controls.
[c]9 missing

**Evaluation of the AI test of specific anti-*Treponema* IgG.** We found that an AI of ≥1.7 as a positive test for NS diagnostic was optimal. At that threshold, 59% of our NS patients had a positive AI.

Regarding NS1, AI was useful for the diagnosis of NS with area under the curve (AUC) of 0.96 (CI 95% of 0.91 to 1.00) (Fig. 1). AI specificity was 98.6% (CI 95% of 92.4 to 100.0) and sensitivity was 81.3% (CI 95% of 54.4 to 96.0). Positive and negative predictive values were >95% for a prevalence above 25% and below 20%, respectively (Fig. 2).

For NS2, AUC was 0.88 (CI 95% of 0.81 to 0.96). AI specificity was still high at 98.6% (CI 95% of 92.4 to 100.0), but sensitivity was reduced to 27.3% (CI 95% of 6.0 to 61.0). Positive and negative predictive values were >95% for a prevalence above 50% and below 6%, respectively.

In NS1 patients, 13/16 had a positive intrathecal synthesis (type II or III OCB). All positive CSF-OCB correlated with a positive specific AI, except in one case with positive CSF-OCB and a negative AI (<1.7), but this intrathecal immune reaction may be triggered by its HIV coinfection.

Among controls, 34/71 (47.9%) had a positive syphilitic serology (TPHA/TPPA of ≥80), but they did not meet the criteria for NS. All patients with other neurological

**TABLE 2** Biological characteristics of CSF

| Variable | NS patients (n = 27) | Control patients (n = 71) | NS1 (n = 16) | NS2 (n = 11) | P value of: | |
|---|---|---|---|---|---|---|
| | | | | | NS vs control | NS1 vs NS2 |
| WBC (cells/mm³) | | | | | <0.0001 | 0.941 |
| Median | 8.0 | 2.0 | 10.0 | 8.0 | | |
| Min–max | 0–110 | 0–190 | 0–110 | 1–79 | | |
| Presence of plasma cells, % (n) | 63% (17) | 27% (19) | 50% (8) | 82% (9) | 0.001 | 0.217 |
| Protein level (g/L) (normal value: 0.15 to 0.45 g/L) | | | | | 0.021 | 0.027 |
| Median | 0.48 | 0.40 | 0.52 | 0.39 | | |
| p25–p75 | 0.36–0.60 | 0.31–0.50 | 0.44–0.74 | 0.32–0.50 | | |
| Albumin quotient >$Q_{Lim}$, % (n) | 26% (7) | 21% (15) | 38% (6) | 9% (1) | 0.611 | 0.183 |
| Total IgG index ≥0.7, % (n) | 33% (9) | 14% (10) | 56% (9) | 0% (0) | 0.031 | 0.003 |
| Intrathecal synthesis, % (n) | 48% (13) | 27% (19) | 81% (13) | 0 | 0.044 | <0.001 |
| Type II OCB, n | 7 | 12 | 7 | 0 | | |
| Type III OCB, n | 6 | 7 | 6 | 0 | | |
| Positive intrathecal anti-*Treponema* IgG antibody index (>1.7), % (n) | 59.3% (16) | 1.4% (1) | 81.3% (13) | 27.3% (3) | <0.001 | 0.015 |

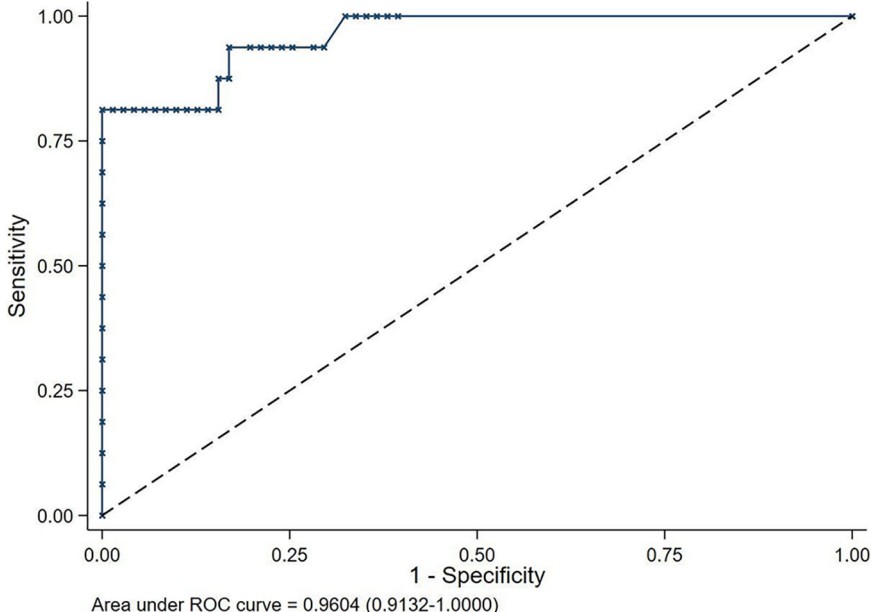

**FIG 1** Sensitivity/(1 − specificity) and AUC of AI test in cases of NS1.

pathologies and a nonreactive treponemal antibody test in the serum had a negative AI (<1.7), except one.

## DISCUSSION

The diagnosis of NS is still an issue and relies on a combination of neurological manifestations in patients with a reactive serological anti-*Treponema pallidum* test and CSF abnormalities. In this study, we investigated the diagnostic performance of CSF-*Treponema pallidum*-specific antibody production by measuring an AI in NS patients classified according to two "gold" standard definitions combining clinical and biological manifestations.

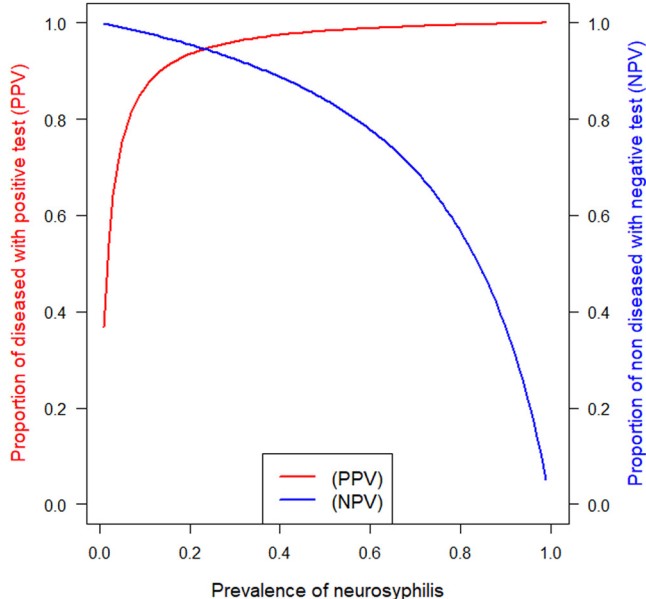

**FIG 2** PPV/PNV of AI test for the diagnostic of NS1 using the optimal threshold >1.7.

In NS1, we found that AI had a high sensitivity and specificity as well as good positive and negative predictive values in the studied population. Although the number of cases included is limited, we believe that this test performs better than others and can be used to diagnose NS alone. Indeed, it reflects a specific intrathecal synthesis by taking into account a possible dysfunction of the BBB, which is not the case when using treponemal tests in the CSF alone, as they can be positive when their blood titer is high or when there is a rupture of the BBB (22). Furthermore, sensitivity of total CSF-IgG synthesis is limited (81.3%) and similar to that of multiple sclerosis (82.3 to 93.0% [23]). Isoelectric focusing tests used to detect OCB in CSF in response to pathogen invasion lack specificity and sensitivity, likely because the limit of detection of IEF is below that of enzyme-linked immunosorbent assay (ELISA). Moreover, OCB can also be detected in CSF of HIV patients (24).

The situation is more complex concerning syphilis with acute ocular or hearing symptoms (NS2). Among the 11 cases analyzed, 3 had no CSF abnormalities, 5 had only high CSF-WBC, and 3 had a CSF-WBC and elevated CSF-protein, but none had a CSF-TPPA of ≥320. For those patients, our AI appears to be helpful to confirm NS diagnosis if positive, but it cannot be used to exclude the diagnosis. This is consistent with a previous study reporting that CSF-*Treponema pallidum* tests suffer from low sensitivity in ocular and otologic NS (25). It is likely that NS2 physiopathology differs substantially from that of NS1. Indeed, NS2 could be considered a syphilis-associated peripheral neuropathy without detectable CSF-*Treponema pallidum* at the time of lumbar puncture. We may assume that *Treponema pallidum* spares the CNS in NS2, at least at initial stages, but we cannot reject a possible delayed CNS invasion. We believe that, when *Treponema pallidum* AI is <1.7 in a patient with ocular or otologic syphilis, a combination of clinical presumptive signs and abnormalities of CSF parameters prevails to establish the diagnosis. In addition, given that mild CSF abnormalities do not have significant impact on treatment strategy (26), we raise the question of the usefulness of lumbar puncture for ocular or otologic syphilis investigations.

This study has several limitations. First, our initial NS1 definition including a CSF-TPPA above or equal to 320 can lead to underdiagnosis of NS. Indeed, it is known that a CSF-TPPA of ≥320 has a low sensitivity (12.8 to 17.4% according to Marra et al. [14]), and therefore, some NS1 cases may be misclassified and AI specificity may be overestimated. Second, because NS is an uncommon disease, multicentric studies are needed to obtain a larger population in order to achieve a higher statistical power to further evaluate AI accuracy, especially in patients with ocular and hearing symptoms, in which CSF abnormalities are less common.

Our data suggest that the lack of specificity of the current NS diagnostic criteria could be improved by intrathecal synthesis index of specific anti-*Treponema pallidum* IgG. However, in otologic or ocular syphilis, with few or no CSF abnormalities, AI is not sufficient to rule out the diagnosis of NS, and further investigations should focus on those cases.

## MATERIALS AND METHODS

**Study design.** We conducted a monocentric diagnostic study with retrospective and prospective recruitment between July 2007 and May 2021 at Geneva University Hospitals, Switzerland.

The study protocol was approved by the Ethics Committee of the Swiss Cantonal Research Ethics Commission (study number 2019-00232).

**Study population and settings.** Eligible patients were all adults presenting neurological and neurosensory signs (ophthalmic and auditory), with serum and CSF sampled less than 3 days apart. CSF samples with red blood cells (RBC) of >200 M/L and albumin quotient of >20 were considered contaminated by blood and were thus further excluded.

**Cases and controls definition.** Cases were patients with NS defined on the basis of composite criteria both clinical (neurological signs) and biological according to European and American guidelines (reference definitions) (2, 7, 8).

Our first composite definition (NS1) includes the following: (i) neurological symptoms or signs consistent with NS, (ii) a positive TPHA/TPPA serology (TPHA/PPA ≥80) and a positive CSF-TPHA/TPPA test of ≥320 (11, 19, 22, 27), and (iii) CSF-WBC of >5 cells/mm³ and/or CSF-protein of >0.45 g/L and/or a reactive CSF-VDRL/RPR test.

CSF examination being normal in 90% of otosyphilis cases and in 30 to 40% of ocular syphilis cases (2, 28), we used a second definition for NS (NS2) based on clinical and serological findings: (i) acute ocular symptoms (recent sudden decrease in visual acuity) and/or acute otologic dysfunctions (sudden hearing loss, acute tinnitus or vertigo) without an alternative diagnosis, (ii) a positive TPHA/TPPA serology (TPHA/TPPA of

≥80), and (iii) a response to NS treatment as assessed by ophthalmologic or hearing tests and at least a 4-fold reduction of RPR/VDRL titer in blood after 12 months.

Controls were patients diagnosed with any other CNS pathologies: infections (neuroborreliosis), inflammatory diseases (multiple sclerosis, Guillain-Barre syndrome), dementia, stroke, acute psychiatric disorders, etc. They could have a positive TPHA/TPPA serology (≥80).

**Data collection and variables.** Patient charts were reviewed and the following data were collected in an online case report form using the Research Electronic Data Capture (REDCap) software: age, gender, neurological symptoms (central and peripheral, otologic and ocular), blood parameters (albumin and total IgG, TPPA/TPHA, RPR/VDRL, HIV infection status, and CD4 number), and CSF results (WBC and RBC counts, cell distribution, total IgG, albumin, OCB distribution patterns, TPPA/TPHA, FTA, and RPR/VDRL).

Albumin and total IgG determination was performed using Atellica Neph630 (Siemens, Zürich, Switzerland). TPPA, TPHA, RPR, and VDRL titrations were performed manually using reagents from Fujirebio, Siemens, Human diagnostic, and Bio-Rad, respectively.

CSF cytological examination consisted of counting WBC and RBC (cells/mm³) and evaluating the proportion of different cell types after CSF cytocentrifugation and May-Gründwald Giemsa staining. The BBB integrity was evaluated according to Reiber method (29). Intrathecal IgG production was qualitatively detected by the gold standard isoelectric focusing (IEF) method (30) according to the instructions of the Hydrasis apparatus supplier (Sebia, Evry, France) and using an IgG-specific antibody staining (Sebia). For those samples, CSF and serum were analyzed on the same gel, and the presence of at least two IgG bands refers to an OCB-positive sample. Intrathecal synthesis of oligoclonal IgG corresponds to two OCB patterns: the type II characterized by the presence of OCB only in CSF and not in serum and the type III characterized by OCB in both CSF and serum with additional bands exclusively in CSF.

**Definition of the index test: intrathecal synthesis index of specific anti-*Treponema* antibodies.** Before testing, prospectively collected serum and CSF samples were stored at 4°C for up to 1 week, while archived retrospective samples were stored at −80°C.

All matched CSF/serum samples were tested in the same series for anti-*Treponema pallidum* IgG with an ELISA containing recombinant *Treponema pallidum* proteins (reference EI2111-9601 G, Euroimmun, Lübeck, Germany). Serial dilutions of CSF (1/6 and 1/36) and serum (1/100, 1/1,000, and 1/6,000) were performed and analyzed with DSX Dynex automate to obtain an optical density (OD) value within the linear range of the test. Only OD values within the linear range of the test were chosen to quantify anti-*Treponema* IgG. Assays were performed by batch between 2018 and 2021.

The AI calculation was determined according to Reiber's method (29). AI is the ratio of Q-trep IgG and Q-total IgG if Q IgG is below $Q_{Lim}IgG$, with $Q_{Lim}IgG = 0.93 \times \sqrt{Qalb^2 + 6 \times 10^{-6}} - 1.7 \times 10^{-3}$, Q-trep IgG = $1,000 \times (CSF\text{-}trep\ IgG_{RLU/mL}/serum\text{-}trep\ IgG_{RLU/mL})$, Q-total IgG = $CSF\text{-}IgG_{mg/L}/serum\text{-}IgG_{g/L}$. Reiber's diagrams for NS1 and NS2 can be found in Fig. S1 in the supplemental material. According to Reiber, a cutoff value of 1.4 should be used, whereas others suggested a cutoff value of 2 or 3 to demonstrate local pathogen-specific antibody production in CSF (30, 31). Thus, we have established an AI cutoff to maximize the diagnostic performance of the test using receiver operating characteristic (ROC) curve.

**Data analysis.** We provide descriptive statistics by the NS definitions (NS1 and NS2) and controls: continuous variables were presented by mean (± standard deviation [SD]) and median (interquartile range), and categorical variables were represented by frequencies and relative percentages. We compared data (age) between NS1 and NS2 and then across the three groups (NS1, NS2, and controls) using Mann-Whitney nonparametric test. We compared HIV status and sex between NS1 and NS2 and then across the three groups (NS1, NS2, and controls) using either chi-squared or Fisher's exact test, depending on the application conditions.

We estimated the area under the curve (AUC) of the AI test with its 95% confidence interval (95% CI) for the diagnosis of NS according to definitions 1 (NS1) and 2 (NS2). We reported the optimal threshold of the AI test that maximized the number of well-classified subjects; we reported the sensitivity, specificity, positive and negative predictive values, positive and negative likelihood ratios, and their 95% CIs using this optimal cutoff. The interpretation of the AI test utility was judged to be highly useful when positive and (negative likelihood)$^{-1}$ ratios were both above 10, moderately useful when they were between 5 and 10, and poorly useful when they were between 2 and 5.

**Data availability.** Raw data can be shared upon request.

## SUPPLEMENTAL MATERIAL

Supplemental material is available online only.
**SUPPLEMENTAL FILE 1**, PDF file, 0.2 MB.

## ACKNOWLEDGMENTS

We thank the technicians in the division of laboratory medicine for their technical help. We thank the Swiss National Fund grant CRSII5 186394 for supporting this project.

All authors declare to have no conflict of interest.

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
