## [Reviewer comments · Microbiology Spectrum]

Microbiology Spectrum

Intrathecal synthesis index of specific anti-treponema IgG: a new tool for the diagnosis of neurosyphilis

Chloe Alberto, Christine Deffert, Nathalie Lambeng, Gautier Breville, Angèle Gayet-Ageron, Patrice Lalive, Laurence Toutous-Trellu, and Lionel Fontao

Corresponding Author(s): Chloe Alberto, Geneva University Hospitals

Review Timeline:

Submission Date:	September 10, 2021
Editorial Decision:	November 27, 2021
Revision Received:	December 20, 2021
Accepted:	January 2, 2022

Editor: Smita Iyer

Reviewer(s): Disclosure of reviewer identity is with reference to reviewer comments included in decision letter(s). The following individuals involved in review of your submission have agreed to reveal their identity: Eman Ahmed El-Attar (Reviewer #3)

Transaction Report:

DOI: <https://doi.org/10.1128/spectrum.01477-21>

November 27, 2021

Dr. Chloe Alberto
Geneva University Hospitals
Division of Dermatology and Venereology
Rue Gabrielle-Perret-Gentil 4
Geneva 1205
Switzerland

Re: Spectrum01477-21 (Intrathecal synthesis index of specific anti-treponema IgG: a new tool for the diagnosis of neurosyphilis)

Dear Dr. Chloe Alberto:

Link Not Available

Sincerely,

Smita Iyer

Journals Department
Reviewer comments:

Reviewer #1 (Comments for the Author):

Ms. No. Spectrum-01477-21, by Alberto and colleagues, introduces a new promising tool highly specific for Neurosyphilis (NS) diagnosis. It is excellent, well-written and it was integrated with all organizational and scientific aspects. However, re-checking the punctuations throughout the manuscript is needed.

Reviewer #2 (Comments for the Author):

In this manuscript, the authors developed a new method for the diagnosis of neurosyphilis. By testing the antibody index (AI) for intrathecal synthesis of specific anti-treponema IgG through ELISA, the authors discovered the correlations of the AI with neurosyphilis in related patients. The data from the manuscript well support the better performance of this method comparing to other known method for the diagnosis of neurosyphilis. While this diagnostic method still suffers some limitations and it is only based on a small number of patients but considering the current lack of efficient diagnostic methods for detecting neurosyphilis

and the challenge for recruiting a large number of patients for this rare disease, the study can still be considered as significant. In future, more related work could be further pursued to improve the method to make it have practical applications. Overall, the content of this manuscript is interesting and can attract readers in the field. However, below listed comments regarding to the manuscript should be addressed.

Major

1. The authors do not provide the resources of reagents or other materials utilized in experiments.
2. The experimental details are not provided in the manuscript.

Reviewer #3 (Public repository details (Required)):

Although data sets are not large, many points will be clarified if raw data is made available

Reviewer #3 (Comments for the Author):

My comments are attached below. Many numbers require clarification

Staff Comments:

Preparing Revision Guidelines

Please return the manuscript within 60 days; if you cannot complete the modification within this time period, please contact me. If you do not wish to modify the manuscript and prefer to submit it to another journal, please notify me of your decision immediately so that the manuscript may be formally withdrawn from consideration by Microbiology Spectrum.

The following points are required to be clarified by the authors:

Line 39 & Line 164: Patients samples were collected over long period of time (almost 15 years). It is not clear in the Methods Section if the samples (CSF & serum) were aliquoted & stored for the period of the study then analysed for anti-Treponema IgG all at the same time or if the samples were tested one by one as soon as the samples were available. If samples were stored, please state the storage conditions & how you assessed if the analyte was stable in the samples for such a prolonged period of time. If the samples were analysed one by one or batch by batch, was the same methodology (ELISA kit) with comparable performance specifications (analytical sensitivity, accuracy, linearity.....) used over 15 years? This is unlikely. Kindly clarify.

Line 42 & Line 372 (Table2): I believe the unit for Protein in CSF is incorrect. Reference range for Protein in CSF is 15-40 mg/dL or 150-400 g/L or 0.15-0.4 mg/mL. Also the unit for WBCs in CSF is cmm. Please correct in the table.

Line 44, Line 142 & Line 372 (Table2): Controls were patients suffering from infectious or inflammatory central nervous system disorders. Would it have been more meaningful if you had chosen patients with negative CSF findings as controls? Also please explain how the protein & leucocyte count in all these controls were within normal range in spite of the inflammatory/infectious state.

Line 52 & & Line 372 (Table2): AI was increased in 16/27 of NS patients (that is 59%) & 13/16 of NS 1 patients (72%). How was the sensitivity of AI calculated as 81.3%. . Also, 1/71 controls (1.4%) had a high AI, so it is not 1%.

Line 47: I would be interesting to add the Reiber diagram & briefly show the calculation of AI.

Line 63: It would be interesting to add more recent references of increase in Syphilis incidence as CDC 2019

Line 89 : Grammar correction: Given these limitations

Line 197 & Line 368 (Table1): It is quite surprising that HIV status was not different between the Syphilitic patients & the controls. Since it is a sexually transmitted disease, it would be comprehensible that STDs are transmitted together. The HIV seropositivity in controls is extremely high (11 patients out of 71) that is 15.5 %. Again, this is calculated in Table 1 as 17.7% which is incorrect. The high seropositivity for HIV in controls must be explained as it is much higher than the general population prevalence.

Line 206 & Line 372 (Table2): How was the large numbers of plasma cells detected in both patients & control groups (27% up to 82%) although the total WBC count is very minimal (2 cells/cmm in controls & 8 cells/cmm in NS patients)

Line 374: It would be interesting to add ROC curves for other tests of NS used in the study as TPPA to compare the performance of all tests. Also, there a spelling mistake is found in the word Sensitivity.

Response to Reviewers

Dear Reviewers,

Thank you very much for considering our work.

Below you can find the answers to the questions point by point.

Modifications in the article are underlined in yellow.

Reviewer #1 (Comments for the Author):

Ms. No. Spectrum-01477-21, by Alberto and colleagues, introduces a new promising tool highly specific for Neurosyphilis (NS) diagnosis. It is excellent, well-written and it was integrated with all organizational and scientific aspects. *However, re-checking the punctuations throughout the manuscript is needed.*

Dear reviewer, thank you very much for your nice comment.

We corrected the punctuations as requested.

Reviewer #2 (Comments for the Author):

In this manuscript, the authors developed a new method for the diagnosis of neurosyphilis. By testing the antibody index (AI) for intrathecal synthesis of specific anti-treponema IgG through ELISA, the authors discovered the correlations of the AI with neurosyphilis in related patients. The data from the manuscript well support the better performance of this method comparing to other known method for the diagnosis of neurosyphilis. While this diagnostic method still suffers some limitations and it is only based on a small number of patients but considering the current lack of efficient diagnostic methods for detecting neurosyphilis and the challenge for recruiting a large number of patients for this rare disease, the study can still be considered as significant. In future, more related work could be further pursued to improve the method to make it have practical applications. Overall, the content of this manuscript is interesting and can attract readers in the field. However, below listed comments regarding to the manuscript should be addressed.

Dear reviewer, thank you for your comment.

We are currently conducting a multicentric cohort study among patients with a suspicion of neurosyphilis followed among 15 hospitals / outpatient clinics in Switzerland with the aim to improve the precision around the diagnostic performance indices of AI.

Below are our answers to your comments.

Major

1. The authors do not provide the resources of reagents or other materials utilized in experiments.
More information are now provided in the materials and methods section (lines 155-158).

2. The experimental details are not provided in the manuscript.

More information about how were performed Trepo IgG and IEF assays are now provided in the materials and methods section (lines 159-166; 173-186).

Reviewer #3 (Public repository details (Required)):

Although data sets are not large, many points will be clarified if raw data is made available

We are conducting currently a larger retrospective and prospective study of patients with suspected neurosyphilis, including some data of this study. We therefore prefer to share our data only on request and not to register our database on a clinical repository.

We added the sentence “Raw data can be shared upon request.”, Line 405.

Reviewer #3 (Comments for the Author):

Line 39 & Line 164: Patients samples were collected over long period of time (almost 15 years). It is not clear in the Methods Section if the samples (CSF & serum) were aliquoted & stored for the period of the study then analysed for anti-Treponema IgG all at the same time or if the samples were tested one by one as soon as the samples were available. If samples were stored, please state the storage conditions & how you assessed if the analyte was stable in the samples for such a prolonged period of time. If the samples were analysed one by one or batch by batch, was the same methodology (ELISA kit) with comparable performance specifications (analytical sensitivity, accuracy, linearity.....) used over 15 years? This is unlikely. Kindly clarify.

We have now better described how samples were stored and used (lines 173-181).

As far as we know, IgG are highly stable molecules that keep their reactivity over long period. Supporting that, our internal quality control for RPR is being using since 2015 without loss of reactivity. Unfortunately, because of the limited volume available, it was not possible to repeat TPPA on archived CSF to assess stability of anti-treponemal Ig.

Line 42 & Line 372 (Table2): I believe the unit for Protein in CSF is incorrect. Reference range for Protein in CSF is 15-40 mg/dL or 150-400 g/L or 0.15-0.4 mg/mL.

Also the unit for WBCs in CSF is cmm. Please correct in the table.

The normal protein range varies among assays used by laboratories but is typically about 15 to 40 milligrams per deciliter (mg/dL) or 0.15 to 0.40 milligrams per milliliter (mg/mL) or **0.15 to 0.40 g per liter (g/L)** (Krieg et Kjeldsberg, Cerebrospinal fluids. In: Henry JB, Ed. Clinical Diagnosis and management by laboratory methods. Philadelphia, Pa: WB Saunders; 1991:445-473).

We changed M/L to cells/mm³ in Table 2.

Line 44, Line 142 & Line 372 (Table2): Controls were patients suffering from infectious or inflammatory central nervous system disorders. Would it have been more meaningful if you had chosen patients with negative CSF findings as controls? Also please explain how the protein & leucocyte count in all these controls were within normal range in spite of the inflammatory/infectious state.

We selected controls presenting the same clinical signs as cases because in routine they would have received the same clinical investigation (index test) performed. Moreover, we wanted to avoid the risk for spectrum bias which tends to overestimate specificity and also global diagnostic accuracy.

Among our controls, we had a mix of patients with both normal and abnormal CSF findings. By selecting only normal CSF findings, our index test would have been automatically negative among controls, leading to an overestimation of its performance and therefore of its specificity. Furthermore, it would not reflect the real-life conditions in which the test would be further performed in routine.

Among our controls, we had mostly CSF exploration for the assessment of paresthesia, peripheral sensory-motor deficits, dementia, confusion, or acute psychiatric disorders among which proteins and leucocytes counts were normal. We also had inflammatory and infectious diseases among which proteins and leucocytes counts were high. However, we provided the results in terms of median and percentile (25p / 75p) because average is sensitive to extreme values, which leads in our case not to be representative of our results.

In the aim to be clearer in the definition for controls, we modified the sentence as follows:

- line 142: "Controls were any patients diagnosed with any other CNS pathology: infections (neuroborreliosis), inflammatory diseases (multiple sclerosis, Guillain-Barre syndrome), dementia, stroke, acute psychiatric disorders, ect".
- Line 44: "Controls were patients with other central nervous system disorders than neurosyphilis".
- And in table 2, we provided the minimum and maximum result values, in place of percentiles.

Line 52 & 218 & Line 372 (Table2): AI was increased in 16/27 of NS patients (that is 59%) & 13/16 of NS 1 patients (72%). How was the sensitivity of AI calculated as 81.3%. Also, 1/71 controls (1.4%) had a high AI, so it is not 1%.

16 NS on 27 patients = 59%

13 AI positive in 16 NS1 patients = 81.25 % (100X13/16)

We estimated the diagnostic performance of AI in both **subgroups, NS1 and NS2.**

Calculation for sensitivity of AI in NS1 = True Positive/(True positive +False Negative) = 13/(13+3) = 0.8125

Results were correct and described accurately in the text: “For NS1, specificity of the IEF was 73.2% (CI95% 61.4–83.1) and sensitivity 81.3% (CI95% 54.4-96.0).”

For controls: 1/71= 1.4%, 1.4% so rounded up to 1% as there were no decimal in our table. We have added a decimal point to all the results to be more precise.

Line 47: I would be interesting to add the Reiber diagram & briefly show the calculation of AI.

We added Reiber diagram of illustrative cases of NS1 and NS2 as supplementary figure 1. The calculation of AI is also now better described in the materials and methods section.

Line 63: It would be interesting to add more recent references of increase in Syphilis incidence as CDC 2019

We added as requested more recent references (CDC 2019, European data 2007-2018, and Ghanem, The modern epidemic of syphilis).

Line 89 : Grammar correction: Given these limitations

Corrected.

Line 197 & Line 368 (Table1): It is quite surprising that HIV status was not different between the Syphilitic patients & the controls. Since it is a sexually transmitted disease, it would be comprehensible that STDs are transmitted together. The HIV seropositivity in controls is extremely high (11 patients out of 71) that is 15.5 %. Again, this is calculated in Table 1 as 17.7% which is incorrect. The high seropositivity for HIV in controls must be explained as it is much higher than the general population prevalence.

High seropositivity for HIV in controls was explained by the study recruitment of patients mostly in infectious diseases wards and this can have considerably increased the numbers of HIV + patients in our control cohort. Indeed, our Hospital is a reference center for HIV infection (www.shcsfoundation.ch).

We thought that it was a strength in our control population to have a high number of HIV + comparable to the NS population because we know that HIV patients have altered CSF parameters and it can be sometimes difficult to differentiate neuro-HIV from neurosyphilis using only CSF routine parameters. Moreover, it makes the two populations comparable.

We added the sentence line 197 (line 214 with corrections): “The high rate of HIV positive patients in the controls is explained by their recruitment mainly from the Infectious diseases department, which is a referral center for HIV infection”.

For table 2, there were missing data on HIV status for 9 controls.

The rate of positives was calculated from the available data, i.e. 11 positives out of 62 = 17.7%.

Line 206 & Line 372 (Table2): How was the large numbers of plasma cells detected in both patients & control groups (27% up to 82%) although the total WBC count is very minimal (2 cells/cmm in controls & 8 cells/cmm in NS patients)

Plasma cell number is not the median of their percentage among all WBC but the number of patients having plasma cells in their CSF, either 1 or more. For example, the number 27% for control patients means that at least one plasma cell was identified in the CSF of 19 control patients among the 71. In addition, we use cytocentrifugation to prepare our cytology slides. This technique allows us to have more than 100 WBC on the slide and detect plasma cells although the total WBC count is very low.

Line 374: It would be interesting to add ROC curves for other tests of NS used in the study as TPPA to compare the performance of all tests. Also, there a spelling mistake is found in the word Sensitivity.

Indeed, it would be interesting to compare the performance of all tests, but from a methodological point of view, there is no sense to compare TPPA or RPR alone to current reference tests as TPPA and RPR are already included in the definition of the reference test. This will violate the diagnostic studies principles on independency between the reference and the index tests (cf. STARD guidelines by Bossuyt PM et al. Ann Intern Med 2003;138:W1-W12).

The spelling mistake in the word "sensitivity" was corrected.

January 2, 2022

Dr. Chloe Alberto
Geneva University Hospitals
Division of Dermatology and Venereology
Rue Gabrielle-Perret-Gentil 4
Geneva 1205
Switzerland

Re: Spectrum01477-21R1 (Intrathecal synthesis index of specific anti-treponema IgG: a new tool for the diagnosis of neurosyphilis)

Dear Dr. Chloe Alberto:

Your manuscript has been accepted, and I am forwarding it to the ASM Journals Department for publication. You will be notified when your proofs are ready to be viewed.

Sincerely,

Smita Iyer
Editor, Microbiology Spectrum

Journals Department
Supplemental Figure 1: Accept